# Inhibition Mechanism of L-Cysteine on Maillard Reaction by Trapping 5-Hydroxymethylfurfural

**DOI:** 10.3390/foods10061391

**Published:** 2021-06-16

**Authors:** Shiqiang Yang, Zhongfei Zhang, Jiaoyong Li, Yuge Niu, Liangli Lucy Yu

**Affiliations:** 1Institute of Food and Nutraceutical Science, School of Agriculture and Biology, Shanghai Jiao Tong University, Shanghai 200240, China; ysqysq@sjtu.edu.cn (S.Y.); 605200122@sjtu.edu.cn (J.L.); 2Shenzhen Key Laboratory of Marine Microbiome Engineering, Institute for Advanced Study, Shenzhen University, Shenzhen 518060, China; zinfly@szu.edu.cn; 3Department of Nutrition and Food Science, University of Maryland, College Park, MD 20742, USA; lyu5@umd.edu

**Keywords:** L-cysteine, Maillard reaction, inhibition, HMF, sulfhydryl compound

## Abstract

The Maillard reaction (MR) can affect the color, flavor, organoleptic properties, and nutritional value of food. Sometimes, MR is undesirable due to lowering the nutrient utilization, producing harmful neo-formed compounds, etc. In this case, it is necessary to control MR. Some chemical substances, such as phenolic acid, vitamins, aminoguanidine, and thiols extracted from garlic or onion, can effectively prevent MR. In this study, L-cysteine (L-cys) was found to inhibit MR after screening 10 sulfhydryl compounds by comparing their ability to mitigate browning. The inhibition mechanism was speculated to be related to the removal of 5-hydroxymethylfurfural (HMF), a key mid-product of MR. The reaction product of HMF and L-cys was identified and named as 1-dicysteinethioacetal–5-hydroxymethylfurfural (DCH) according to the mass spectrum and nuclear magnetic resonance spectrum of the main product. Furthermore, DCH was detected in the glutamic–fructose mixture after L-cys was added. In addition, the production of DCH also increased with the addition of L-cys. It also was worth noting that DCH showed no cell toxicity to RAW 264.7 cells. Moreover, the in vitro assays indicated that DCH had anti-inflammatory and antioxidant activities. In conclusion, L-cys inhibits MR by converting HMF into another adduct DCH with higher safety and health benefits. L-cys has the potential to be applied as an inhibitor to prevent MR during food processing and storage.

## 1. Introduction

The Maillard reaction (MR) is a kind of non-enzymatic browning reaction that occurs between the amino group of amino acids and the carbonyl group of reducing sugars during food processing and storage. It shows both desirable and undesirable effects within this process. MR produces many flavor compounds such as aldehydes, ketones, pyridines, and pyrazine, which form the special flavor of some foods [1]. Especially for coffee and bread, MR provides tantalizing aroma and color to increase customers’ appetites. However, sometimes they are undesirable in dairy and other foods [2,3]. The brown pigment is also produced during MR in some foods, such as dried fruit, fruit juice, milk, honey, etc., which makes them unacceptable and cuts down their market value [4,5,6,7]. In addition, the reaction reduces the nutritional value of foods by changing the structure of amino acids and sugars in them [8].

N-glucosamine is first formed when carbonyl compounds react with amino compounds in MR, and it subsequently transforms to 5-hydroxymethylfurfural (HMF), finally, HMF continues to react with amino compound to generate advanced glycosylation end products (AGEs) and melanoidins. Many toxic chemicals, such as glyoxal, methylglyoxal, heterocyclic amines, and acrylamides are produced during this chain reaction [9]. As a key intermediate of MR, HMF is a kind of food contaminant with noticed cytotoxicity [10]. HMF and its two oligomers exert neurotoxic effects in vivo and in vitro [11]. HMF also can be metabolized into other toxic compounds in the body, such as 5-sulfoxymethyfurfural (5-SMF) [12], 5-chloromethylfurfural (5-CMF) [13], and 2,5-dimethylfuran (2,5-DMF) [14], which have carcinogenic, mutagenic, and organ toxicities [15,16]. Almost all industrial foods, such as cereals, dried jams, honey, beverages, milk, and condiments, contain HMF [17]. Even in some coffee and cookies, the concentration of HMF can exceed 1000 mg/kg [18,19], and long-term intake of HMF causes damage to the human body [20,21]. AGEs, downstream products of HMF, are associated with diabetes, kidney disorder, cardiovascular disease, aging, and Alzheimer’s disease [12].

Therefore, it is necessary to reduce food contaminants derived from MR. Some methods have been proposed; for example, temperature, pH, water activity, and other food processing and storage conditions were selected as control points to regulate MR [5]. The substrates of MR, mainly reducing sugars and amines, can be selected to control the reactivity [22,23,24,25]. What is more, some chemical compounds were added into food to block reactive sites, intermediates, or products to suppress MR, such as aminoguanidine, epicatechin, tea polyphenol, phloretin and phloridzin, quercetin, pyridoxamine, stilbene glucoside, ferulic acid, etc. [26]. Among these compounds, sulfur dioxide, thiols, onion extracts, and garlic extracts exhibit a certain ability to inhibit MR [27], and the possible mechanism is that the sulfhydryl group scavenges the reactive carbonyl group to inhibit MR [28].

In this work, we aim to investigate whether sulfhydryl compounds have the potential to be developed as MR inhibitors by deciphering the possible mechanism. Ten sulfhydryl compounds (1-mercaptoglycerol, 3-mercapto-1-propanol, 3-mercaptopropanoic acid, 3-mercaptopropionic acid methyl ester, (2R)-2-amino-3-mercaptopropanoic acid, 3-mercapto-2-butanone, 2-mercapto-3-butanol, 2-mercaptosuccinic acid, 1,3-dimercaptopropane, and 1,6-dimercaptohexane) from food additives were first screened to select candidate MR inhibitor by comparing their inhibitory effects on the model MR, which consisted of fructose and glutamic acid to simulate the processing of fruit juice [29]. The compound possessing the best inhibitory effect to the model MR was chosen as the candidate MR inhibitor for further research. On one hand, a possible mechanism by which sulfhydryl compound inhibits MR was proposed through the characterization of the product of sulfhydryl compound and HMF, the key intermediate in MR. On the other hand, the safety of the candidate MR inhibitor was evaluated by comparing the biological activity (cell toxicity, anti-inflammatory activity, extracellular antioxidant activity, and intracellular antioxidant activity) of HMF and the product.

## 2. Materials and Methods

### 2.1. Chemicals and Reagents

The HMF, TiCl_4_, D-fructose, 1-mercaptoglycerol (1-M-G), 3-mercaptopropanoic acid (3-MPA), 3-mercaptopropionic acid methyl ester (3-MPAME), (2R)-2-amino-3-mercaptopropanoic acid (L-cys), 1,3-dimercaptopropane (1,3-DMP), 1,6-dimercaptohexane (1,6-DMH), and dimethyl sulfoxide (DMSO) were purchased from Aladdin Reagents Database Inc. (Shanghai, China). The L-glutamic acid; 2, 2-diphenyl-1-picrylhydrazyl radical (DPPH**•**); 2, 2-azinobis (3-ethylbenzothiazoline-6-sulfonic acid) diammonium salt (ABTS**•**); lipopolysaccharide (LPS); thiazolyl blue tetrazolium bromide (MTT); and 6-hydroxy-2,5,7,8-tetramethylchroman-2-carboxylic acid (Trolox) were purchased from Sigma-Aldrich (Shanghai) Trading Co., Ltd. (Shanghai, China). The acetonitrile, methanol, 3-mercapto-2-butanone (3-M-2-Butanone), and 2-mercapto-3-butanol (2-M-3-Butanol) were purchased from Adamas Reagent Co., Ltd. (Shanghai, China). The 3-mercapto-1-propanol (3-M-1-P) and 2-mercaptosuccinic acid (2-M-SA) were purchased from Sarn Chemical Technology Co., Ltd. (Shanghai, China). 2,2-azobis (2-amidinopropane) dihydrochloride (AAPH) was purchased from J&K Scientific Co., Ltd. (Shanghai, China). Other reagents were purchased from Sinopharm Chemical Reagent Co., Ltd. (Shanghai, China).

### 2.2. Inhibitory Effect of Sulfhydryl Compounds on MR

Fructose (carbonyl donor) and glutamic acid (amino donor) were dissolved in phosphate buffer (75 mM, pH 7.4) to obtain the model MR. The stock solution (100 mM) of sulfhydryl compounds was prepared using phosphate buffer (75 mM, pH 7.4). Then, 4.9 mL of the fructose–glutamic acid solution and 0.1 mL of stock solution were mixed in the test tube to obtain a 5 mL reaction system (the final concentration of fructose–glutamic acid was 0.2 M). Then, 0.1 mL of phosphate buffer without the sulfhydryl compound was added into 4.9 mL of the fructose–glutamic acid solution as control. The mixtures were then stirred for 9 h by water bath at 80 °C (C-Mag HS 7, IKA Instrument Equipment Co., Ltd., Staufen im Breisgau, Germany) to initiate the model MR. Subsequently, the absorbance value of the reaction solution was measured at 420 nm by a microplate reader (BioTek Instruments, Inc., Winooski, VT, USA) to indicate the degree of browning [30]. The concentration of HMF in the reaction solution was measured by the followed high-performance liquid chromatographic method with diode-array detection (HPLC-DAD method). According to the concentration of the standard HMF solution and corresponding integral area at 284 nm, a calibration curve within the range of 0.015625–2.0 mg/mL was built for quantitative determination of HMF.

### 2.3. HPLC-DAD

A high-performance liquid chromatograph (HPLC, Agilent 1260, Agilent, Palo Alto, CA, USA) with a diode array detector was used for the detection of HMF and the adduct. An Agilent Eclipse XBD-C18 column (4.6 × 250 mm, 5 μm) was used, and the column temperature was 35 °C. The injection volume was 5 μL, and the mobile phase was ammonium formate buffer (30 mM, pH 2.44) (mobile phase A) and acetonitrile (mobile phase B). The gradient elution procedure is shown in Appendix A. In addition, 235 nm, 254 nm, and 284 nm channels were selected to detect HMF and the adduct. The full wavelength scan mode was on.

### 2.4. Preparation of the Adduct of HMF and L-Cys

HMF (10 mmol) and L-cys (25 mmol) were dissolved in 50 mL of acetonitrile in a round-bottom flask and kept in a refrigerator at −15 °C for 5 min. Then, 1.5 mmol of TiCl_4_ was added slowly as the catalyst, and the flask was put back in the refrigerator at −15 °C for another 5 min. Finally, the mixture was reacted at room temperature for 9 h with constant stirring at 180 rpm.

The reaction mixture was first filtrated using qualitative filter paper (15–20 μm) to remove precipitate. Then it was purified by macroporous resin HP-20 (Diaion, Tokyo, Japan; matrix: styrene-divinylbenzene; density: 1.01 g/mL; mean pore size: 260 Å; 4.2 cm i.d. × 15.0 cm; ultrapure water as eluent). The HPLC-DAD method outlined in Section 2.3 was used to determine the fractions in which the product was distributed. The fractions containing the product were collected and concentrated by rotary evaporation (BUCHI, Flawil, Switzerland). Then, Sephadex LH-20 (GE Healthcare, Uppsala, Sweden; granule size: 27–163 μm; 3.6 cm i.d. × 172.5 cm; 5% (*v/v*) methanol as eluent) was used to further purify the product. The eluents containing high-purity product determined by HPLC-DAD (the retention time of the main product, 1-dicysteinethioacetal–5-hydroxymethylfurfural (DCH), was about 5.2 min, as shown in Appendix A) were collected and then lyophilized using a vacuum freeze-dryer (Labconco, Kansas, MO, USA) to obtain the adduct.

### 2.5. UPLC-Q-TOF-MS for DCH Identification

The adduct was dissolved with ultrapure water and analyzed by ultra-performance liquid chromatography-quadrupole time-of-flight mass spectrometry (UPLC-Q-TOF-MS, Waters, Milford, MA, USA). An Acquity UPLC BEH C18 column (2.1 mm i.d. × 100 mm, 1.7 μm) attached to a Van Guard precolumn (2.1 mm i.d. × 5 mm, 1.7 μm) (Waters, Milford, MA, USA) was used. The condition was shown as: column temperature, 40 °C; mobile phase A, ultrapure water; mobile phase B, acetonitrile (MS grade); 0.1% (*v/v*) formic acid was added in mobile phases A and B; injection volume, 2 μL. The gradient elution procedure is shown in Appendix A.

For mass spectrometry, the electrospray ionization (ESI) source was used in negative ion mode. The sodium formate (LC-MS grade, 0.34 g/L in 90% isopropanol, *v/v*) was used to calibrate the mass axis, the range of which was 100–1800 (*m/z*); leucine-enkephalin (LE) was used for real-time calibration, the concentration of which was 2 ng/L with a flow rate of 10 μL/min. The molecular weight [M+H]^+1^ was 556.2771. We used a capillary voltage of 2.5 kV, a sampling cone voltage of 35.0 kV, an extraction cone voltage of 3.0 kV, a source temperature of 120 °C, a desolvation temperature of 450 °C, a cone gas flow rate of 50.0 L/h, and a desolvation gas flow rate of 600.0 L/h. Two channels of mass spectrometry measurements were recorded. The first channel collected data of mass spectrometry signals ranging from 100–1800 Da with a 6 eV collision energy. The second channel offered additional fragmentation information for the compound identification, which collected mass fragments of 100–1800 Da with a 30 eV collision energy.

### 2.6. Nuclear Magnetic Resonance (NMR)

The adduct (3.5 mg) was transferred to the nuclear magnetic resonance tube (WG-10000-7, Wilmad-LabGlass, NJ, USA) and was fully dissolved in 1.0 mL D_2_O (Adamas, Shanghai, China). The 1D and 2D NMR analyses were carried out on a Bruker Avance III 600 MHz NMR spectrometer (Bruker, Rheinstetten, Germany) equipped with a ^1^H/^13^C/^15^N 5 mm TCl CryoProbe at 25 °C. The chemical shift was expressed in ppm, and TMS was used as an internal reference (0 ppm).

### 2.7. Detection of DCH during MR

To verify that L-cys could inhibit MR by trapping HMF through thioacetal reaction, different concentrations of L-cys were added to the fructose–glutamic acid reaction system, and the corresponding DCH concentrations were detected by HPLC-DAD. First, 4.9 mL of the fructose–glutamic acid solution and 0.1 mL of L-cys solution were mixed to obtain a 5 mL reaction system (fructose–glutamic acid concentration: 0.2 M; L-cys concentrations: 50, 100, or 200 mM) before heating. The mixture was stirred for 9 h at 80 °C in the water bath. For quantitating DCH in the reaction solution, a series of standard DCH solutions (0, 200, 500, 1000, 5000 mg/L) was prepared first. Their integral areas at 254 nm were obtained according to the method given in Section 2.3. Based on the concentration of the standard DCH solution and corresponding integral area at 254 nm, a calibration curve within the range of 0–5000 mg/L was built for quantitative determination of DCH. The purity of standard was confirmed by the ^1^H NMR spectrum of DCH (Appendix A).

### 2.8. Cytotoxicity and Biological Activity of DCH in RAW 264.7 Cells

#### 2.8.1. Cell Culture

The murine monocyte/macrophage cell line RAW 264.7 was obtained from the Chinese Academy of Sciences Shanghai Cell Bank (Shanghai, China). The cells were cultured in a 100 mm Nunc EasYDish round cell culture dish (Thermo Fisher, Waltham, MA, USA) in Dulbecco’s modified Eagle’s medium (DMEM) supplemented with 10% FBS and 1% penicillin–streptomycin under 5% CO_2_ at 37 °C. Cells were passed every 3 days for experiments. The RAW 264.7 cells were detached by a scraper, centrifuged, and then plated in a fresh medium for continued growth.

#### 2.8.2. Cytotoxicity Assay

The MTT assay was performed according to the procedure reported by Mossmann [31] with slight modifications. To measure the cytotoxicity of HMF and the adduct in cell proliferation, 100 μL of RAW 264.7 cell suspension (1 × 10^5^ cells/mL) were seeded into a 96-well plate in triplicate and pre-incubated for 24 h for cell adherence. The culture medium was removed, and 100 μL of fresh medium including HMF or DCH was added. The concentrations of HMF or DCH were prepared as 80, 160, and 320 μM. The cell was incubated under 5% CO_2_ at 37 °C for another 20 h. Subsequently, 10 μL of tetrazolium salt solution (5 mg/mL MTT in Phosphate Buffered Saline (PBS)) was added to each well in the dark. Four hours later, the medium was removed and 200 μL of DMSO was added to dissolve the formazan crystals, followed by shaking for 10 min. The absorbance of samples was measured at 570 nm. The cell viability ratio (%) compared with the blank group was calculated using the following equation:(1)$$\mathrm{Cell}\;\mathrm{viability}\;\left(\%\right)=\frac{\mathrm{absorbance}\;\mathrm{of}\;\mathrm{test}\;\mathrm{sample}}{\mathrm{absorbance}\;\mathrm{of}\;\mathrm{blank}}\;\times \;100\%$$

#### 2.8.3. Anti-Inflammatory Activity

To compare the anti-inflammatory activity of HMF and the adduct DCH, Nitric oxide (NO) production in LPS-stimulated RAW 264.7 cells was examined. The NO determination method was used according to the previous reference, with slight modifications [32]. RAW 246.7 cells were seeded in 96-well plates with a density of 1 × 10^5^ cells/well and grown for 12 h in adherence. The cells were treated with test samples for 1 h and then incubated for 24 h in fresh DMEM with or without 1 μg/mL of LPS. The nitrite concentration in the culture medium was measured as an indicator of NO production according to the procedure of the NO Test Kit (production number S0021, Beyotime Biotechnology, Shanghai, China). Briefly, 50 μL of the cell culture supernatant and nitrite solution (100, 50, 25, 12.5, 6.25, 3.125, and 0 μM) were seeded in a 96-well plate, and then 50 μL of Griess reagent I and 50 μL of Griess reagent II was added to each well. After reacting for 5 min, the absorbance was recorded at 540 nm using the microplate reader (BioTek Instruments, Inc., Winooski, VT, USA). At the same time, the cell viability was measured by an MTT assay. The NO concentration produced by unit cells (ratio of NO concentration to cell activity) was used to represent the level of inflammation accurately.

The pro-inflammatory factors, including tumor necrosis factor (TNF)-α, and interleukin (IL)-1β and IL-6, were examined to investigate the anti-inflammatory activity of DCH. RAW 264.7 cells were incubated for 24 h at 37 °C to reach a confluence of 80%. The cells were pretreated with HMF or DCH (100, 200, and 300 μM) for 1 h and then incubated for 4 h in fresh DMEM with or without 1 μg/mL of LPS. Culture supernatant was collected to determine the content of protein production using the commercial kits (eBioscience; Thermo Fisher Scientific Inc., Waltham, MA, USA; cat. nos. BMS607-3, BMS6002, and BMS603-2).

#### 2.8.4. Relieving H_2_O_2_-Induced Oxidative Stress in RAW 264.7 Cells

Cell viability after H_2_O_2_ inducing was measured to compare intracellular antioxidant activity according to a previous report with minor modifications [33]. Briefly, RAW 264.7 cells were seeded in a 96-well plate with a density of 1 × 10^5^ cells/mL for 12 h for adherence. At the end of the cultivation time, the medium was changed to DMEM with 100 μL of HMF or DCH for another 12 h. The H_2_O_2_ (500 μM) was then added to induce cell apoptosis for 4 h. The medium was removed again, and the fresh medium was added with tetrazolium salt solutions to each well in the dark. After the formazan crystals were fully formed, the medium was discarded and 200 μL of DMSO was added to dissolve the formazan crystals. After shaking for 10 min, absorbance was measured at 570 nm. The cell viability ratio (%) was calculated using the following equation:(2)$$\mathrm{viability}\;\mathrm{ratio}\;\left(\%\right)=\frac{\mathrm{absorbance}\;\mathrm{of}\;\mathrm{test}\;\mathrm{sample}}{\mathrm{absorbance}\;\mathrm{of}\;\mathrm{control}}\times 100\%$$

### 2.9. Free Radical Scavenging Ability

#### 2.9.1. Relative DPPH Radical Scavenging Capacity (RDSC)

The comparison of the relative DPPH**•** scavenging capacity between HMF and DCH was conducted according to a laboratory protocol reported previously [34]. First, 100 μL of the sample, Trolox standard solution, or blank solution (methanol, HPLC-grade) was pipetted into appropriate wells. Then, 100 μL of 0.2 mM DPPH**•** solution (dissolved in methanol) was sequentially added to initiate the radical–antioxidant reaction. The absorbance at 515 nm was recorded every minute for 1.5 h using a Synergy 2 multi-mode microplate reader (BioTek, Winooski, VT, USA). DPPH**•** (%) quenched for all standards and samples at all time points was calculated using the following equation:(3)$$\mathrm{DPPH}•\;\mathrm{quenched}\;\mathrm{at}\;t\min\left(\%\right)=\left(1-\frac{A_{sample}-A_{blank}}{A_{control}-A_{blank}}\right)\times 100\%$$
where *A_sample_*, *A_blank_*, and *A_control_* represent the absorbance of the sample, blank, and control wells, respectively, at 515 nm at *t* min.

After obtaining the graphs of the changes in absorbance values of samples, blanks, and controls over time, the relative DPPH**•** scavenging capacity was calculated based on the area under the curve (AUC) and reported as micromoles of Trolox equivalents (TE) per gram of sample.

#### 2.9.2. ABTS Cation Radical (ABTS**•**^+^) Scavenging Capacity Assay (ASCA)

The free radical scavenging capacities of HMF and DCH were evaluated against ABTS**•**^+^ generated according to previously reported protocols [35]. The ABTS**•**^+^ radicals stock solution was generated by oxidizing a 5 mM aqueous solution of ABTS**•** with manganese dioxide under ambient temperature for 30 min and then diluting it with 1.5 mM phosphate buffer to form a working solution with an absorbance of 0.7 at 734 nm. Then, the mixture was added with 1 mL of ABTS**•**^+^ working solution and 80 µL standard (Trolox solution of different concentrations) or sample solution in a centrifuge tube to react for 90 s. The mixture’s absorbance was then determined at 734 nm. Methanol was used as the blank. Trolox equivalents per gram of sample were calculated using a standard curve.

#### 2.9.3. Oxygen Radical Absorbance Capacity (ORAC)

The ORAC assay was carried out according to a laboratory protocol reported previously [35]. First, 30 μL of HMF, DCH, Trolox standard solution, or methanol (blank control) was mixed with 225 μL of 81.6 nM fluorescein (FL), which was used as a molecular probe. After 20 min of preheating at 37 °C in a Synergy 2 multi-mode microplate reader, 25 μL of AAPH solution (0.36 M) was successively added into each well to initiate the reaction. The fluorescence was recorded every minute for 2 h at 37 °C. Excitation and emission wavelengths were 485 nm and 528 nm, respectively. The ORAC value was calculated from the integral area under the curve of the graph of fluorescence intensity over time and was expressed as micromoles of Trolox equivalents (TE) per gram of sample. All the tests were performed in triplicate.

### 2.10. Statistical Analysis

Data are reported in mean ± standard deviation (SD) in triplicate. Statistical significance was declared at *p* < 0.05 followed by one-way analysis of variance (ANOVA). An S−N−K post hoc test was used to identify differences between groups in all assays. All statistical analyses were performed using SPSS software (Version 23.0, SPSS, Inc., Chicago, IL, USA).

## 3. Results and Discussion

### 3.1. Inhibitory Effect of Sulfhydryl Compounds on MR

The glutamic acid and fructose were chosen to react to simulate MR. The absorbance at 420 nm and the concentration of HMF in reaction liquid were determined to show the inhibitory effect of 10 sulfhydryl compounds as food additives, including 1-M-G, 3-M-1-P, 3-MPA, 3-MPAME, L-cys, 3-M-2-butanone, 2-M-3-butanol, 2-M-SA, 1,3-DMP, and 1,6-DMH (Appendix A). The results are shown in Figure 1.

The data of the blank group indicated that d-fructose and l-glutamic acid led to Maillard browning and generated 2.47 mM of HMF at certain reactive conditions. Compared with the blank group, the absorbance value and HMF content of samples added with 1-M-G, 3-M-1-P, 3-MPA, 3-MPAME, and L-cys were lower. The 1-M-G, 3-M-1-P, 3-MPA, 3-MPAME, and L-cys decreased the absorbance value by 56.6%, 49.3%, 49.3%, 38.8%, and 66.4%, respectively; and the HMF content by 48.7%, 48.4%, 53.8%, 40.3%, and 96.3%, respectively. This indicated that these sulfhydryl compounds reduced the degree of Maillard browning. The reaction solution added with 2-M-3-butanol and 1,3-DMP had no difference in browning degree compared with the blank group. However, the addition of 1,3-DMP increased the concentration of HMF by 80.7% in the reaction solution compared with the blank group. The addition of 3-M-2-butanone, 2-M-SA, and 1,6-DMH also increased the browning degree (208.0%, 125.9%, and 190.4%, respectively) and HMF content (347.5%, 242.2%, and 503.1%, respectively) of the reaction solution compared with the blank group.

The valence electron of the sulfur atom on the mercapto group is far away from the nucleus. The degree of polarization of the sulfur atom is larger than the oxygen atom. So, it is easy for sulfur atoms to give electrons and exhibit strong nucleophilicity. At the same time, the oxygen atom of the HMF aldehyde group has a high electronegativity, which makes the carbon atom positively charged [36]. Thus, the mercapto group very easily attacks the aldehyde group carbon atom to undergo a nucleophilic addition reaction to form a mercaptal, thereby reducing the content of HMF. Moreover, the aldehyde group of HMF can also react with an amino to form a Schiff base [37]. The furan ring of HMF can react with amino and sulfhydryl groups through a Michael-type addition [38]. In this study, 1-MG, 3-M-1-P, 3-MPA, 3-MPAE, and L-cys of the three-carbon skeleton had the ability to inhibit Maillard browning and reduce the content of HMF in the reaction system. Considering that HMF is an important intermediate in the browning process, these compounds may react with HMF to prevent the process of HMF from producing brown pigments.

The structures of 1-MG and 3-M-1-P were similar (Appendix A), and 1-MG had just one more hydroxyl group than 3-M-1-P. Their inhibitory effects on the degree of Maillard browning and HMF production were also similar, indicating that the hydroxyl group hardly affected the inhibitory effect on browning and HMF content, because the polarization of the hydroxyl group was not as strong as that of the mercapto group. Furthermore, 3-MPAME and L-cys are compounds obtained from 3-MPA by substitution of different groups at different positions. It is seen in Figure 1 that the amino substitute showed better performance in controlling Maillard browning than the methyl substitute. The amino group may react with the aldehyde group and furan ring of HMF to form Schiff bases and Michael adducts to eliminate HMF [37,38].

The 3-M-2-butanone, 2-M-3-butanol, and 2-MSA were all mono-sulfhydryl compounds with a four-carbon skeleton. They did not effectively inhibit Maillard browning and HMF production compared to mono-sulfhydryl compounds with a three-carbon skeleton, and even promoted the production of HMF (3-M-2-butanone and 2-MSA). Although 1,3-DMP and 1,6-DMH are compounds containing two mercapto groups, they did not show the inhibitory effect of browning. The 1,6-DMH even aggravated the Maillard browning, the mechanism of which was unclear. Among these compounds that prevented Maillard browning, L-cys showed the best effect. The inhibition mechanism of L-cys on MR by trapping HMF was then investigated.

### 3.2. Structural Identification of the Adduct of HMF and L-Cys

The main product of the reaction between L-cys and HMF was obtained by a purification using macroporous resin HP-20 and Dextran gel LH-20 to remove the impurities (Appendix A). The molecular weight of the product showed an ion peak *m/z* 349.0534 [M-H]^−^ in the HR-ESI-MS (calculated for C_12_H_17_N_2_O_6_S_2_, *m/z* = 349.0528) (Appendix A).

In the ^1^H spectra of the product (Appendix A and Table 1), the two signals at 6.39 ppm and 6.51 ppm indicated that there was only one hydrogen nucleus on their adjacent carbon atoms. In addition, the integral value indicated that there was also only one hydrogen atom connected to the carbon atoms. These two signals matched with the two hydrogen nuclei from the furan ring of HMF. The peak with δH of 4.55 ppm was a single peak, indicating that there was no hydrogen nucleus on the adjacent carbon atom. Meanwhile, the peak integral area was 2H, which suggested that this signal peak was generated by two hydrogen atoms in the methyl group. The signal peak of the hydrogen atom from the aldehyde group in HMF disappeared in favor of a new signal peak with δH of 5.37 ppm. This signal value was consistent with the hydrogen atom on the thioacetal structure. In addition, the nuclear magnetic hydrogen spectrum signal values of the two groups of L-cys appeared in the spectrum.

Combined with the results of mass spectrometry and NMR spectrum, we inferred that the product was an adduct obtained by the thioacetal reaction of one molecule of HMF with two molecules of L-cys (Figure 2a), which was further verified by the NMR carbon spectrum (Appendix A). The structure of the adduct is shown in Figure 2b; it was named as 1-dicysteinethioacetal–5-hydroxymethylfurfural (DCH).

### 3.3. Confirmation of the Generation of DCH during MR Added with L-Cys

According to Figure 3, DCH was detected in the fructose–glutamic acid reaction solution by adding L-Cys. In the concentration range of 50–200 mM, the more L-cys was added, the more DCH was generated. This proved the mechanism that L-cys inhibited MR by trapping HMF to produce the adduct DCH. Meanwhile, the mechanism may also be applicable to other sulfhydryl inhibitors.

### 3.4. The Cytotoxicity of HMF and DCH on RAW 264.7 Cells

This experiment aimed to measure the cytotoxicity of HMF and DCH on RAW 264.7 cells. Toxicity results for different concentrations of HMF and DCH on RAW 264.7 cells are shown in Figure 4a. Compared with the blank group, HMF and DCH in the concentration range of 80–320 μM did not decrease the cell viability significantly, which meant there was no obvious cytotoxicity of HMF or DCH.

A previous study conducted by Zhao Q [39] reported that HMF expressed significant injury to Caco-2 cells, and the cell viability decreased with an increase in the concentration of HMF. However, the cell type and the concentration of HMF and DCH were different from ours. In Zhao Q’s study, a concentration range of 8 mM to 128 mM was used, which was too high above the actual intake concentration to cause cell death in our opinion, as the maximum concentration in our study was only 320 μM. Moreover, in addition to being a sulfhydryl compound, L-cys is also an amino acid, and since it was reported that the formation of 5-hydroxymethylfurfural-lysine Schiff base significantly decreased the cytotoxicity of HMF against GES-1 cells, EA.hy926 cells, and Caco-2 cells [40], we supposed that L-cys may reduce cytotoxicity of HMF on RAW 264.7 cells by forming mercaptal with HMF when the concentration is high, which needs further verification.

### 3.5. Anti-Inflammatory Activity Comparison between HMF and DCH

The NO produced per unit cell (NPC) in each experimental group is shown in Figure 4b. The NPC of the blank group was very low, which was the value under normal physiological conditions. When LPS was added, the inflammatory pathway was activated and a large amount of NO was produced, about 10 times the NPC under normal biological conditions. The NPC of groups added with HMF or DCH was lower than that of the LPS group, indicating that HMF and DCH inhibited the production of NO.

Above 80 μM, the NPC of the DCH experimental groups was significantly lower than that of the HMF experimental groups, showing that L-cys transformed HMF into a compound with better anti-inflammatory activity. The aldehyde group of HMF was eliminated by a thioacetal reaction between L-cys; furthermore, two sulfur atoms, two carboxyls, and two amino groups were introduced into the molecule. This structural change led to the enhancement of the anti-inflammatory activity of DCH. The possible principle can be deepened by exploring the effects of two compounds, HMF and DCH, on the activity of nitric oxide synthase or the expression of nitric oxide synthase [41].

The secreted protein levels of IL-6, IL-1β, and TNF-α were measured using the medium of DCH-pretreated RAW 264.7 cells. As shown in Figure 4c, DCH decreased the production of IL-6 slightly in a dose-independent manner. However, the content of IL-1β and TNF-α in the culture medium was not detected.

Macrophages are a type of inflammatory cell that play an important role in the initiation and development of the inflammatory process and the stimulation of the production of pro-inflammatory mediators. Activated macrophages produce a series of pro-inflammatory mediators, including nitric oxide (NO), which plays an important role in the development of various chronic diseases [42]. Given that NO content is highly correlated with inflammation levels in the body, the content of NO is often used as an indicator of the level of inflammation in the disease process. Therefore, L-cys converted HMF into DCH, not only mitigating the degree of Maillard browning, but also heightening the anti-inflammatory effect.

### 3.6. Extracellular Antioxidant Activity Comparison between HMF and DCH

The free radical scavenging abilities of HMF and DCH measured using the three methods of RDSC, ASCA, and ORAC are listed in Table 2. The scavenging ability for three kinds of free radicals per gram of HMF was low (equivalent to the scavenging ability of Trolox of 215.14, 293.92, and −439.04 μmol, respectively), and the value in the ORAC experiment was lower than zero, indicating that HMF did not slow the recession of fluorescence in the ORAC experiment; on the contrary, it promoted the oxidation of the fluorescent probe and accelerated the decrease of the fluorescence intensity in the solution. The free radical scavenging ability per gram of DCH was much higher, about 10 times that of HMF (equivalent to the scavenging ability of Trolox of 2110.04, 2942.47, and 3260.25 μmol, respectively). The results of different free radical scavenging experiments on the same substance may have been different, but there was no doubt that DCH had better antioxidant ability than HMF.

The principle of the RDSC and ASCA experiments was that free radicals in the form of single electrons exhibit a dark color that will disappear when the reducing substance provides electrons for their electronic pairing. There is a furan ring in HMF, and the double bond and the oxygen atom of the furan ring in HMF do not give electrons, but share electrons with single-electron atoms, which also fades the color of free radicals. While the reducibility exhibited in this way was not strong, HMF only showed weak free radical scavenging ability. DCH also has a furan ring, which reveals reducibility in the same way as HMF. However, the thioacetal structure of DCH was untied and released mercapto groups to display the strong reducibility. In addition, the original two carboxyl groups of L-cys also had reducibility, making the free radical scavenging ability of DCH far exceed that of HMF.

The free radicals used in RDSC (DPPH**•**) and ASCA (ABTS**•**^+^) were organic nitrogen-centered free radicals, while the single-electron nitrogen in the DPPH**•** free radical was surrounded by three benzene rings. Thus, the steric hindrance during RDSC was great, causing difficulty in reducing the DPPH**•**. The principle used in the ORAC (ROO•) experiment was different from that of the RDSC and ASCA experiments. It relied on the transfer of hydrogen atoms to quench the peroxy radicals. The aldehyde group of HMF also competed for the hydrogen atoms on the fluorescent probe to quench the probe, and accelerated the attenuation of the fluorescence intensity, so the result for the ORAC experiment was negative. The aldehyde group in DCH was destroyed, and mercapto groups dissociated from the thioacetal. The mercapto groups and carboxyl groups of L-cys reacted with the peroxy radicals to provide hydrogen atoms to the peroxy radicals and quenched the radicals so that the fluorescent probes remained, and the fluorescence decline slowed. Prior pointed out that the free radicals used in the ORAC experiment; that is, peroxy radicals, were biologically relevant free radicals, rather than artificial free radicals. Hence, the free radical scavenging ability detected by the ORAC experiment reflected the actual condition in the organism more than the RDSC and ASCA experiments [43].

### 3.7. Intracellular Antioxidant Activity Comparison

Figure 4d shows the cell viability under different concentrations of HMF and DCH. Compared with the blank group, H_2_O_2_ induced oxidative damage to cells, resulting in a decrease in the cell survival rate of more than 60%. In the concentrations of 80–320 μM, HMF reduced the cell survival rate significantly, more than 80% compared with the blank group. According to the result, HMF higher than 80 μM even aggravated the oxidative damage of cells, while DCH alleviated the decrease in the cell survival rate caused by the oxidative damage of hydrogen peroxide in this concentration range.

The oxidative damage experiment of cells better reflected the actual free radical scavenging ability of the compound in the organism. This experiment and the ORAC experiment yielded the same result—not only did HMF have no ability to scavenge the free radical, but also it exacerbated oxidative damage. As a kind of α, β-unsaturated aldehyde, the aldehyde group of HMF has electrophilicity, which attacked proteins, nucleic acids, and lipids when ingested in the body, causing damage and an increase in free radicals in the body. In addition, the results of this experiment and the free radical scavenging experiment showed that DCH possessed excellent antioxidant capacity and exhibited better biological activity than HMF.

What is more, the results of the antioxidant experiment and the anti-inflammation experiment were consistent. Oxidative stress and inflammation are associated with the process of many diseases, such as diabetes, obesity, cardiovascular disease, cancer, etc. [44,45,46,47,48,49,50]. The increase of free radicals in the body easily causes proteins, cell membrane and nucleic acids to be attacked, and triggers inflammation. Meanwhile, the free radicals, such as active oxygen radicals, act as second messengers to activate the release of inflammatory factors [51,52]; therefore, HMF may have a pro-inflammatory effect. Similarly, DCH has an outstanding free radical scavenging ability, thereby reducing the inflammatory response in the body and reducing the release of NO inflammatory factors.

## 4. Conclusions

MR has positive and negative effects in different situations and different foods. For some foods, inhibitors of MR were found to control the disadvantage of MR. We found several sulfhydryl compounds, including L-cys, that inhibited the degree of browning and the content of HMF effectively. The mechanism of L-cys was also proposed as: L-Cys formed an adduct DCH with HMF to reduce the content of HMF and blocked the downstream reaction in MR. According to the results of the biological activities of HMF and DCH, L-cys converted HMF into DCH with higher anti-inflammatory and antioxidant activities, while reducing the toxic effects of HMF. The result indicated that L-cys was safe and effective as an MR inhibitor. It can be applied in food processing and storage to reduce the content of the food contaminant HMF. Adding L-cys will provide healthier food and beverages, especially dried fruit, fruit juice, dairy, and honey. Furthermore, there is a need to evaluate the organoleptic properties and flavors of foods after adding such inhibitors.

## Figures and Tables

**Figure 1 foods-10-01391-f001:**
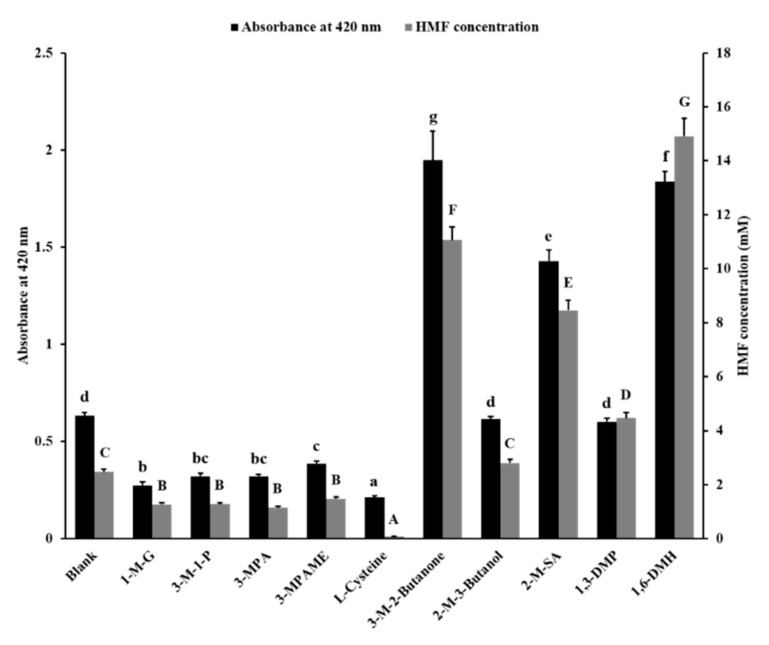
Inhibitory effect of sulfhydryl compounds on Maillard browning and 5-hydroxymethylfurfural (HMF) concentration in the fructose–glutamic acid model system. Different letters indicate a significant difference, *p* < 0.05. 1-M-G, 1-mercaptoglycerol; 3-M-1-P, 3-mercapto-1-propanol; 3-MPA, 3-mercaptopropanoic acid; 3-MPAME, 3-mercaptopropionic acid methyl ester; L-cysteine, (2R)-2-amino-3-mercaptopropanoic acid; 3-M-2-Butanone, 3-mercapto-2-butanone; 2-M-3-Butanol, 2-mercapto-3-butanol; 2-M-SA, 2-mercaptosuccinic acid; 1,3-DMP, 1,3-dimercaptopropane; 1,6-DMH, 1,6-dimercaptohexane.

**Figure 2 foods-10-01391-f002:**
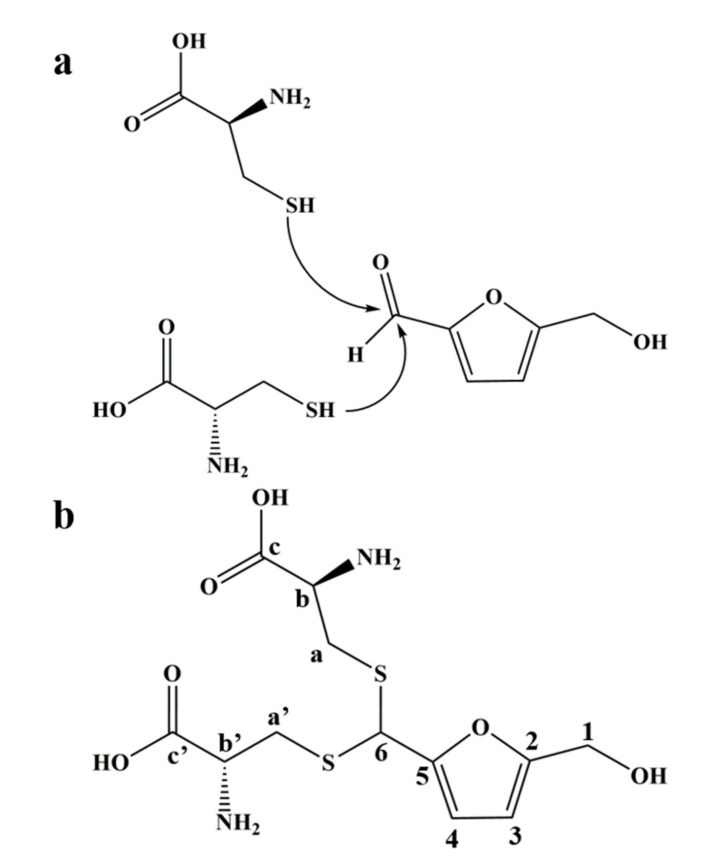
The reaction mechanism and the adduct structure of HMF and L-cys. (**a**) The sulfhydryl group of L-cys nucleophilically attacks the aldehyde carbon of HMF to cause a thioacetal reaction. (**b**) The structure of 1-dicysteinethioacetal–5-hydroxymethylfurfural (DCH) and label of carbon atoms.

**Figure 3 foods-10-01391-f003:**
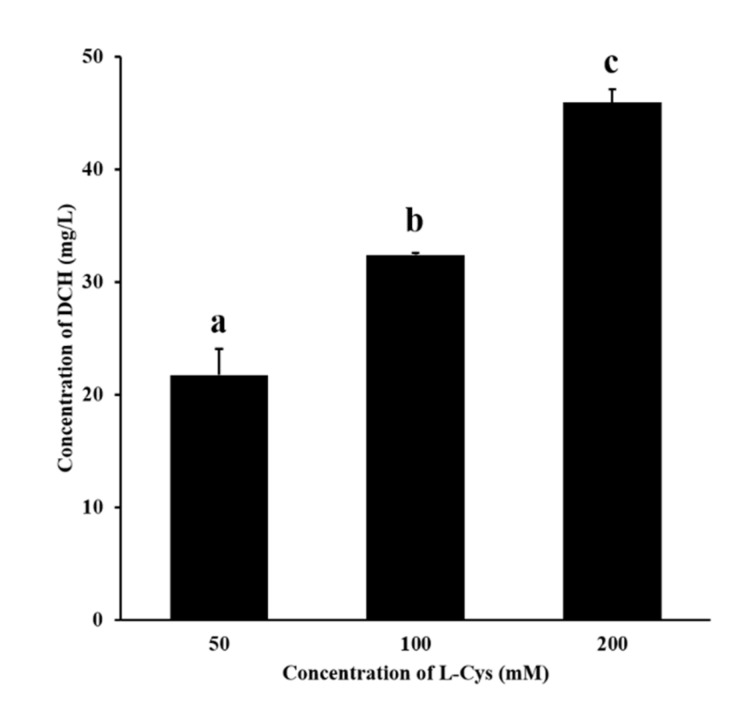
Effect of different concentrations of L-cys on the production of DCH. Different letters indicate significant differences, *p* < 0.05.

**Figure 4 foods-10-01391-f004:**
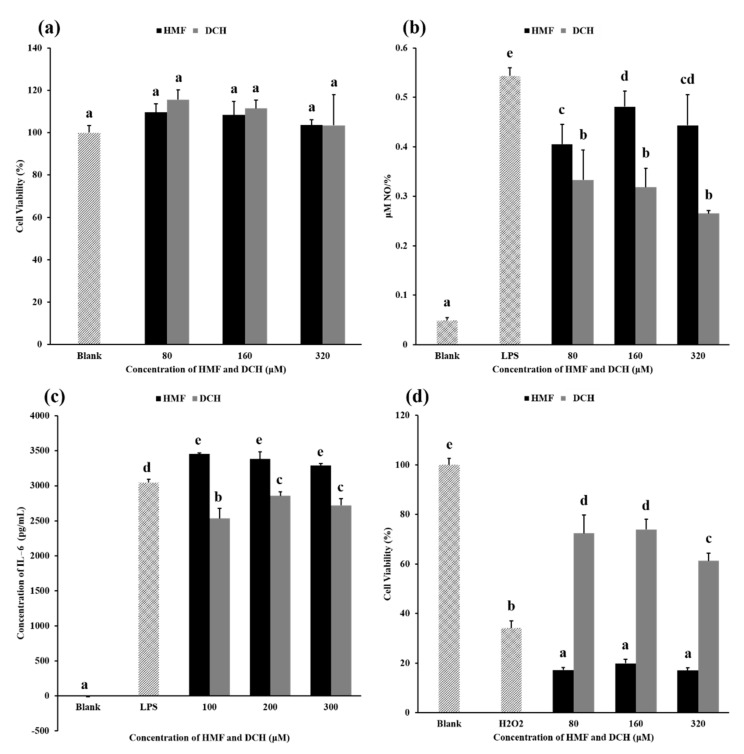
Comparison of biological activity between HMF and DCH. (**a**) The influence of HMF and DCH on the viability of RAW 264.7 cells. (**b**) Effect of HMF and DCH on the Nitric oxide (NO) production in LPS-stimulated RAW 264.7 cells. (**c**) Effects of HMF and DCH on the secreted protein levels of IL-6 in RAW 264.7 cells. (**d**) Cell viability after oxidative damage induced by hydrogen peroxide. Different letters indicate significant differences, *p* < 0.05.

**Table 1 foods-10-01391-t001:** ^1^H NMR (600 MHz) and ^13^C NMR (151 MHz) spectrum data.

Carbon Label *	δH (ppm)	δC (ppm)
1	4.55 (s, 2H)	55.67 (s)
2		154.45 (s)
3	6.51 (d, J = 3.2 Hz, 1H)	109.30 (s)
4	6.39 (d, J = 3.2 Hz, 1H)	110.32 (s)
5		149.62 (s)
6	5.37 (s, 1H)	45.09 (s)
a	3.17 (dd, J =14.7, 7.7 Hz, 1H) 3.06 (dd, J = 14.9, 7.5 Hz, 1H)	32.51 (s)
b	3.82 (ddd, J = 11.6, 7.6, 4.4 Hz, 1H)	53.79 (s)
c		172.39 (s)
a’	3.28–3.19 (m, 2H)	32.04 (s)
b’	3.82 (ddd, J = 11.6, 7.6, 4.4 Hz, 1H)	53.57 (s)
c’	4.55 (s, 2H)	172.43 (s)

* Carbon label is according to Figure 2b.

**Table 2 foods-10-01391-t002:** Results of free radical scavenging experiments.

Experiment Name (Free Radical Type)	RDSC (DPPH•) *	ASCA (ABTS•^+^) *	ORAC (ROO•) *
HMF	215.14 ± 24.17	293.92 ± 81.91	−439.04 ± 119.17 ^a^
DCH	2110.04 ± 100.80	2924.47 ± 70.05	3260.25 ± 97.20

2,2-diphenyl-1-picrylhydrazyl radical (DPPH**•**) was used in relative DPPH radical scavenging capacity experiment (RDSC). 2,2-azinobis (3-ethylbenzothiazoline-6-sulfonic acid) diammonium salt (ABTS**•**) was used in ABTS Cation Radical (ABTS**•**^+^) Scavenging Capacity Assay (ASCA). Oxygen radical (ROO**•**) generated from 2,2-azobis (2-amidinopropane) dihydrochloride (AAPH) was used in oxygen radical absorbance capacity experiment (ORAC). Data are expressed in mean value ± SD (*n* = 3, μmol Trolox equivalent per gram sample). *: The free radical scavenging abilities of HMF and DCH were significantly different, *p* < 0.05. ^a^: The value of scavenging ability of HMF to peroxyl radicals was negative.

## Data Availability

The data presented in this study are available upon request from the corresponding author.

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
