# Peer review of "Inhibition Mechanism of L-Cysteine on Maillard Reaction by Trapping 5-Hydroxymethylfurfural"

_foods, 2021, doi:10.3390/foods10061391_

Round 1
Reviewer 1 Report
The manuscript written by Yang and colleagues is describing the benefits of trapping HMF by L-cysteine forming thus forming an adduct called DTH. This adduct seems to have anti-inflammatory effects. While this approach may be promising in limiting the toxic effects of HMF, a Maillard Reaction Product many issues need to be addressed. Furthermore, from the abstract and during the overall manuscript, authors consider that Maillard Reaction Products form a family of toxic compounds that need to be avoided in foods; This is a very narrow point of view when considering the Maillard Reaction. Indeed, they never consider the benefits of Maillard Reactions products in terms of colour and flavour of foods. This need to be changed to better reflect the reality of this reaction.
Major comments:
Introduction:
Line 34-35. Please change the sentence in order to better reflect the positive and the negative influence of Maillard Reaction.
It sounds that authors consider that they are the first ones to identify this molecule they call DTH. However, they can’t present it as it is since this compounds, 1-Dicysteinethioacetal–5-Hydroxymethylfurfural, has also been described as DCH by Zhao et al (. Agric. Food Chem. 2018, 66, 43, 11451–11458).
Methods:
Part 2.7: Authors say that they detected DTH in the samples. They shall indicate how they proceeded to this detection. Furthermore, when considering the results (Figure 3), we can observe that they give them as a concentration in mg/L. However, this seems to be difficult to proceed without having a patented standard.
Why choosing to work with an adherent immune cell line as they are not on first line of contact with the adduct coming from food. Furthermore, why working with murine cells?
The cytotoxicity measurement (part 2.8.2) is not clearly described. In fact, the authors mention that they expose the cells to MTT before the scheduled culture time came. To which schedule do they refer to? How long did this step last? Furthermore, the timepoint of DTH administration shall be clarified.
Authors worked at a initial cell density of 105 cells/mL while in the literature others work at 106 cells/mL. Could they explain why?
Authors chose to measure as a marker of anti-inflammatory activity (part 2.8.3) nitrite concentration in the culture as a marker of NO production. However, they could have proceeded more easily by measuring, in the medium, pro-inflammatory cytokine secretion (Il-1, Il-6 and/or TNF).
Could the authors justify the fact that they measured antixoxydant activity with 3 different techniques and split them in two separate paragraph they entitled antioxidant activity and scavenging capacity? This is indeed important to clarify since, from the results it seems that some of the techniques seem not to be so appropriate due the experimental design?
Please rephrase the statistical analysis paragraph.
Results:
Why naming their adduct DTH and not DCH as previously described?
How do the authors explain a cell viability level higher than 100% in the cells treated by HMF and DTH?
Conclusion:
The first sentence is not appropriate. This is not quite true since this is not the first time this approach was used.
Minot comments:
Abstract: authors use the name DTH in the abstract but never describe it.
Line 39-40: replace “glycosylation” but “glycation”. Furthermore, to better reflect the chronology of the reaction, authors should present AGEs before melanoidins.
Author Response
Itemized list of responses to reviewers’ comments
Comments of Reviewer 1:
- 1. Introduction: Line 34-35. Please change the sentence in order to better reflect the positive and the negative influence of Maillard Reaction.
Response: Thank for your suggestion. The Maillard Reaction has the positive and the negative influence during food processing and storage. In some cases, Maillard Reaction needs to be controlled, which is the purpose of our research. We have rewritten these sentences in Line 36-39.
- 2. It sounds that authors consider that they are the first ones to identify this molecule they call DTH. However, they can’t present it as it is since this compounds, 1-Dicysteinethioacetal–5-Hydroxymethylfurfural, has also been described as DCH by Zhao et al (. Agric. Food Chem. 2018, 66, 43, 11451–11458).
Response: Thank you for your comment. We identified the product, 1-Dicysteinethioacetal–5-Hydroxymethylfurfural before the publication of Zhao’s paper in July of 2018. At that time, we named 1-Dicysteinethioacetal–5-Hydroxymethylfurfural as DTH and used DTH to refer to the compound throughout the following experiments and drawings. In the process of writing the manuscript, for convenience, we did not change DTH to DCH. According to your suggestion, we now change DTH to DCH in our manuscript.
- Part 2.7: Authors say that they detected DTH in the samples. They shall indicate how they proceeded to this detection. Furthermore, when considering the results (Figure 3), we can observe that they give them as a concentration in mg/L. However, this seems to be difficult to proceed without having a patented standard.
Response: Thank you for your comment. There was no commercial DCH standard. So, we prepared standard DCH with ideal purity (>99%). Then the calibration curve was built based on the concentration of standard DCH solution and corresponding integral area (R2 value as 0.9992). The concentration of DCH in reaction solution was calculated using this calibration curve. We described how we quantitate DCH in Line 174-178.
- Why choosing to work with an adherent immune cell line as they are not on first line of contact with the adduct coming from food. Furthermore, why working with murine cells?
Response: Thank you for your comment. The RAW 264.7 cell is a classic cell line which is used to analyze the anti-inflammatory ability of compounds. The reason was presented in Line 403-409.
- The cytotoxicity measurement (part 2.8.2) is not clearly described. In fact, the authors mention that they expose the cells to MTT before the scheduled culture time came. To which schedule do they refer to? How long did this step last? Furthermore, the timepoint of DTH administration shall be clarified.
Response: Thank you for your suggestion. The scheduled time was 48 h for whole cell culture. The cell was exposed to MTT for 4 h. We have described the detailed procedures on how to conduct the cytotoxicity measurement in Line 190-198.
- Authors worked at a initial cell density of 105cells/mL while in the literature others work at 106 cells/mL. Could they explain why?
Response: Thank you for the comment. The cell density in different literature is different. We decided the proper cell density of 1×105 cells/mL according to preliminary experiment and related reference (Biochemical Pharmacology 2019, 166, 231-241).
- Authors chose to measure as a marker of anti-inflammatory activity (part 2.8.3) nitrite concentration in the culture as a marker of NO production. However, they could have proceeded more easily by measuring, in the medium, pro-inflammatory cytokine secretion (Il-1, Il-6 and/or TNF).
Response: Thank you for your comment. We previously investigated the anti-inflammatory effect of HMF and DCH by measuring TNF-α, IL-6 and IL-1β. The results had no significant difference and concentration-dependence. So we did not show these data in our manuscript.
- Could the authors justify the fact that they measured antixoxydant activity with 3 different techniques and split them in two separate paragraph they entitled antioxidant activity and scavenging capacity? This is indeed important to clarify since, from the results it seems that some of the techniques seem not to be so appropriate due the experimental design?
Response: Thank you for your comment. We have changed the title “antioxidant activity” to “Relieving on H2O2-induced oxidative stress in RAW 264.7 cells” according to other reviewer’s suggestion in Line 217. Both the extracellular free radical scavenging experiment and the intracellular oxidative stress experiment are important to reflect the antioxidant activity of HMF and DCH. Some literatures also investigated the antioxidant activity using different free radicals (Journal of Agricultural & Food Chemistry 2005, 53, 6649; Food Chemistry 2019, 286, 8-16.).
- Please rephrase the statistical analysis paragraph.
Response: Thank you for your suggestion. We have rephrased these sentences in Line 268-272.
- Why naming their adduct DTH and not DCH as previously described?
Response: Thank you for the comment. We have named 1-Dicysteinethioacetal–5-Hydroxymethylfurfural as DCH in the manuscript.
- How do the authors explain a cell viability level higher than 100% in the cells treated by HMF and DTH?
Response: Thank you for the comment. HMF and DCH with certain concentration may slightly promote cell growth. But there is not statistical significance compared with the blank group.
- Conclusion: The first sentence is not appropriate. This is not quite true since this is not the first time this approach was used.
Response: “Firstly” in this sentence did not indicated that we are the first ones to explore structure-effect relationship of sulfhydryl compounds on Maillard Reaction in the world. It means that this is the first step of our research. Thank you for your suggestion. It was revised in Line 488-490.
- Abstract: authors use the name DTH in the abstract but never describe it.
Response: Thank you for the comment. It was revised in Line 20-21.
- Line 39-40: replace “glycosylation” but “glycation”. Furthermore, to better reflect the chronology of the reaction, authors should present AGEs before melanoidins.
Response: Thank you for your suggestion. We have revised it in Line 43-45.
Reviewer 2 Report
Specific Criticisms, Comments, Suggestions:
The present manuscript deals with the capability of different sulfhydryl compounds to mitigate the development of the Maillard reaction, with especial focus on L-cysteine, which showed the higher ability to decrease the presence of HMF. An inhibition mechanism based on a Michael-type addition between L-cysteine and HMF is proposed. Paper is interesting although not very innovative. The manuscript need to be really improved regarding to the English language (revision by a native English speaker), structure and other aspects, some of them are described below:
Abstract
It would be interesting if some data (figures) could be introduced. This usually increases visibility and citation of the paper.
- Line 22:” in vitro assay” instead of “assay in vitro”.
- Line 24: “applied” instead of “developed”.
- Lines 24-25: “L-Cys has the potential to be developed as an inhibitor to prevent MR during food processing and storage”. Authors should consider that this amino acid will contribute with flavour and aroma, so the final food product could have unexpected organoleptic characteristics. A tasting panel should be proper.
Introduction
- Lines 33-35: “The brown pigment is also produced during the Maillard reaction, which makes food unacceptable and cuts down the market value” Authors should take care with this type of affirmations since melanoidins belong to those brown pigments and, for example, they are partially responsible for the organoleptic properties of coffee as well as different biological activities.
- Lines 48-51: “Even in some coffee and cookies, the concentration of HMF can exceed 1000 mg/kg [13, 14] and the daily intake of HMF is many orders of magnitude higher than that of another food toxic substance acrylamide, which exceed the safety threshold, therefore, long-term INTAKE of HMF COULD CAUSE damage to the human body”. This sentence is confusing. Are authors talking about a threshold for HMF or for acrylamide? There’s an important different. In Europe and some other countries there is a regulation with maximum recommended acrylamide levels but nothing exits for HMF except in honey. If authors refer to a threshold for HMF, the regulation must be quoted. Clarify this, please.
- Line 57: Modify like this, please “Besides, the substrates of the Maillard reaction, MAINLY reducing sugars and amines, CAN BE SELECTED TO CONTROL the reactivity.”
- Lines 66-77: this paragraph should describe the aim of the study and how it was performed. Results and conclusions must be deleted in the introduction.
Material and methods
- Lines 94-104: very confusing description of the preparation of the model system, difficult to understand. Re-write it, please.
- Lines 112-115: Replace by “A calibration curve within the range 1.0 mg/mL-0.015625 mg/ml was built for quantitative determination of HMF”. Include here the limit of quantification for your method.
- Consider if section 2.4 could be moved to 2.3 and conversely, since it seems more logical to prepare the adduct and them perform the HPLC and UPLC determinations.
- Lines 122-132: the description of the purification of the adduct should be improved.
- Section 2.4 and 2.5: Authors mention in line 125 they measured the content of the adduct in the eluents by HPLC-DAD. Afterwards, they describe in section 2.5 they measured it by UPLC-Q-TOF-MS. However, I understand that in the second case it was an identification rather than a quantification. Is it right? The purpose of each analysis should be clarified in the text.
- Line 163: “L-Cys with different concentrations WAS added” instead of “concentration WERE”.
- Section 2.7: Authors should clarify if L-Cys was added to the fructose-glutamic acid model system before (still without HMF formed) or after heating (HMF already present in the solution).
- Line 184: include the various concentrations of test samples added.
- Section 2.8.4: The title of the section (Antioxidant activity) is confusing and does not reflect properly the assay performed. I suggest something like “Cell antioxidant protection after oxidative damage”
- Section 2.9.2 and 3, lines 238 and 249: describe concisely the samples for this assay, please.
- Line 271: authors should mention the composition of the blank.
- Lines 279-280: “However, the addition of 1,3-DMP increased the concentration of HMF by 80.7 % in the reaction solution” Compared with??? Clarify, please.
- Results and discussion: I encourage authors the reading of the paper “Formation and elimination reactions of 5-hydroxymethylfurfural during in vitro digestion of biscuits” by HamzalıoÄŸlu and Gökmen (Food Res. Int., 99, 308-314, 2017) because will give interesting information on the matter.
- Lines 303-305: “Considering that HMF…” To understand the sentence, it is necessary to clarify if sulfhydryl compounds were added before or after the heating of the model system. Clarify in the corresponding section of Material and Methods, please.
- Lines 313-315: “The possible reason is that the amino group condenses with aldehyde groups of HMF to form 314 Schiff bases to eliminate HMF”. See also HamzalıoÄŸlu and Gökmen to support this hypothesis.
- Lines 323-324: “The inhibition mechanism by trapping HMF was then investigated on the derivative with its safety”. Confusing sentence, please re-write.
- Consider moving section 3.7 after 3.5.
- Lines 431-432: “The free radicals used in RDSC and ASCA were organic nitrogen-centered free radicals, while the single-electron nitrogen in DPPH· free radical was surrounded by three benzene rings”. Does not refer RDSC and DPPH to the same antioxidant method?? Clarify, please.
- Line 442: Delete initials R L, please.
- Line 469: “Oxidative stress and inflammation are pertinent in the body.” What do the authors mean with this sentence? If they refer both phenomena inevitably occur in the body, it is a rare way of saying it. Clarify, please.
- Conclusions: the whole paragraph has bad writing and many sentences are not adequate for a conclusion. Authors should consider here if the use of L-Cys could be suitable in all food matrices taking into account the possible organoleptic consequences.
Figures and Tables
- Caption of Figure 1: Inhibitory effect of sulfhydryl compounds on Maillard browning and HMF concentration IN THE FRUCTOSE-GLUTAMIC ACID MODEL SYSTEM.
Authors should mention here the composition of the Blank.
- Table 2: It would be clearer if DPPH and ABTS are used instead of RDSC and ASCA, since these are the names more extended for this antioxidant methods. If only figures with the letter a are expressed as μmol Trolox equivalent per gram of sample, which is the unit for the rest of data? Please, consider revision.
Finally, I encourage authors to perform a revision of the whole manuscript by a native English speaker. Several parts are not well understood and it could be due to the language.
Author Response
Itemized list of responses to reviewers’ comments
Comments of Reviewer 2:
- It would be interesting if some data (figures) could be introduced. This usually increases visibility and citation of the paper.
Response: Thank you for your suggestion. We have prepared the graphical abstract and will upload it.
- Line 22:” in vitro assay” instead of “assay in vitro”.
Response: Thank you for your suggestion. We have revised it in Line 25.
- Line 24: “applied” instead of “developed”.
Response: Thank you for your suggestion. We have revised it in Line 27.
- Lines 24-25: “L-Cys has the potential to be developed as an inhibitor to prevent MR during food processing and storage”. Authors should consider that this amino acid will contribute with flavour and aroma, so the final food product could have unexpected organoleptic characteristics. A tasting panel should be proper.
Response: Thank you for your suggestion. We will conduct experiments to investigate it in the future. We added “Furthermore, it needs to evaluate the organoleptic property and flavor of food by adding such inhibitors.” in the conclusion part in Line 497-499.
- Lines 33-35: “The brown pigment is also produced during the Maillard reaction, which makes food unacceptable and cuts down the market value” Authors should take care with this type of affirmations since melanoidins belong to those brown pigments and, for example, they are partially responsible for the organoleptic properties of coffee as well as different biological activities.
Response: Thank you for the comment. We stressed that the brown pigment has different influence in different foods in Line 36-39.
- Lines 48-51: “Even in some coffee and cookies, the concentration of HMF can exceed 1000 mg/kg [13, 14] and the daily intake of HMF is many orders of magnitude higher than that of another food toxic substance acrylamide, which exceed the safety threshold, therefore, long-term INTAKE of HMF COULD CAUSE damage to the human body”. This sentence is confusing. Are authors talking about a threshold for HMF or for acrylamide? There’s an important different. In Europe and some other countries there is a regulation with maximum recommended acrylamide levels but nothing exits for HMF except in honey. If authors refer to a threshold for HMF, the regulation must be quoted. Clarify this, please.
Response: Thank you for the comment. Due to the lack of clinical trial data, so far, there is no regulation about the threshold of HMF. We tried to point out that the intake of HMF is much higher than that of acrylamide to show that HMF is harmful to the human body. According to your comment, we agree it is inappropriate to compare the toxicity of HMF and acrylamide. We delete “the daily intake of HMF is many orders of magnitude higher than that of another food toxic substance acrylamide, which exceed the safety threshold” in the introduction.
- Line 57: Modify like this, please “Besides, the substrates of the Maillard reaction, MAINLY reducing sugars and amines, CAN BE SELECTED TO CONTROL the reactivity.”
Response: Revised it in Line 59-61.
- Lines 66-77: this paragraph should describe the aim of the study and how it was performed. Results and conclusions must be deleted in the introduction.
Response: Revised it in Line 68-82.
- Lines 94-104: very confusing description of the preparation of the model system, difficult to understand. Re-write it, please.
Response: Revised it in Line 100-106.
- Lines 112-115: Replace by “A calibration curve within the range 1.0 mg/mL-0.015625 mg/ml was built for quantitative determination of HMF”. Include here the limit of quantification for your method.
Response: Thank you for your comment. We here made a mistake. The range of the calibration curves was 2.0 mg/mL-0.015625 mg/ml with the R2 value as 0.9983. It was revised in Line 111-113.
- Consider if section 2.4 could be moved to 2.3 and conversely, since it seems more logical to prepare the adduct and them perform the HPLC and UPLC determinations.
Response: Thank you for the comment. We firstly established the HPLC-DAD method to detect whether a product was formed. When confirmed, we began to prepare a large amount of product. We believed it is better to keep the original order of 2.3 and 2.4.
- Lines 122-132: the description of the purification of the adduct should be improved.
Response: Thank you for your suggestion. We have revised it in Line 128-138.
- Section 2.4 and 2.5: Authors mention in line 125 they measured the content of the adduct in the eluents by HPLC-DAD. Afterwards, they describe in section 2.5 they measured it by UPLC-Q-TOF-MS. However, I understand that in the second case it was an identification rather than a quantification. Is it right? The purpose of each analysis should be clarified in the text.
Response: HPLC-DAD was used for quantification of HMF and DCH. UPLC-Q-TOF-MS was used for structure identification. We have clarified the purpose of section 2.5 in Line 139.
- Section 2.7: Authors should clarify if L-Cys was added to the fructose-glutamic acid model system before (still without HMF formed) or after heating (HMF already present in the solution).
Response: Thank you for the comment. L-Cys was added to the fructose-glutamic acid model system before heating. We have revised it in Line 173.
- Line 184: include the various concentrations of test samples added.
Response: Revised it in Line 193-194.
- Section 2.8.4: The title of the section (Antioxidant activity) is confusing and does not reflect properly the assay performed. I suggest something like “Cell antioxidant protection after oxidative damage”
Response: Thank you for your suggestion. We have revised it in Line 217.
- Section 2.9.2 and 3, lines 238 and 249: describe concisely the samples for this assay, please.
Response: Thank you for your suggestion. We have revised it in Line 247 and Line 258-259.
- Line 271: authors should mention the composition of the blank.
Response: Thank you for your suggestion. The blank was composed of PBS (solvent) and we have mentioned it in Line 105-106.
- Lines 279-280: “However, the addition of 1,3-DMP increased the concentration of HMF by 80.7 % in the reaction solution” Compared with??? Clarify, please.
Response: Thank you for your suggestion. The addition of 1,3-DMP increased the concentration of HMF by 80.7 % in the reaction solution compared with the blank group. We have revised it in Line 288-290.
- Results and discussion: I encourage authors the reading of the paper “Formation and elimination reactions of 5-hydroxymethylfurfural during in vitro digestion of biscuits” by HamzalıoÄŸlu and Gökmen (Food Res. Int., 99, 308-314, 2017) because will give interesting information on the matter.
Response: Thank you for your recommendation. It is a useful literature. We inserted it as reference [38] in Line 308-309.
- Lines 303-305: “Considering that HMF…” To understand the sentence, it is necessary to clarify if sulfhydryl compounds were added before or after the heating of the model system. Clarify in the corresponding section of Material and Methods, please.
Response: Thank you for your suggestion. Sulfhydryl compounds were added before heating. We described the detailed procedure in Line 102-104.
- Lines 313-315: “The possible reason is that the amino group condenses with aldehyde groups of HMF to form 314 Schiff bases to eliminate HMF”. See also HamzalıoÄŸlu and Gökmen to support this hypothesis.
Response: Thank you for your suggestion. We discussed the result of this paper in Line 322-324.
- Lines 323-324: “The inhibition mechanism by trapping HMF was then investigated on the derivative with its safety”. Confusing sentence, please re-write.
Response: Thank you for your suggestion. We revised it in Line 332-333.
- Consider moving section 3.7 after 3.5.
Response: The section 3.6 was about the extracellular antioxidant activity while the section 3.6 was about the intracellular antioxidant activity. We believe it is logical to discuss the antioxidant activities from extracellular to intracellular experiments.
- Lines 431-432: “The free radicals used in RDSC and ASCA were organic nitrogen-centered free radicals, while the single-electron nitrogen in DPPH· free radical was surrounded by three benzene rings”. Does not refer RDSC and DPPH to the same antioxidant method?? Clarify, please.
Response: We have revised it in Line 440-442.
- Line 442: Delete initials R L, please.
Response: We have revised it in Line 451.
- Line 469: “Oxidative stress and inflammation are pertinent in the body.” What do the authors mean with this sentence? If they refer both phenomena inevitably occur in the body, it is a rare way of saying it. Clarify, please.
Response: Thank you for the comment. We have revised it in Line 478-480.
- Conclusions: the whole paragraph has bad writing and many sentences are not adequate for a conclusion. Authors should consider here if the use of L-Cys could be suitable in all food matrices taking into account the possible organoleptic consequences.
Response: Thank you for your suggestion. We have rewritten the conclusion part. The organoleptic evaluation will be analyzed in the future. The prospect was added in Line 497-499.
- Caption of Figure 1: Inhibitory effect of sulfhydryl compounds on Maillard browning and HMF concentration IN THE FRUCTOSE-GLUTAMIC ACID MODEL SYSTEM.
Response: We have revised it in Line 299.
- Authors should mention here the composition of the Blank.
Response: Thank you for the comment. We mentioned the composition of the blank it in Line 105-106.
- Table 2: It would be clearer if DPPH and ABTS are used instead of RDSC and ASCA, since these are the names more extended for this antioxidant methods. If only figures with the letter a are expressed as μmol Trolox equivalent per gram of sample, which is the unit for the rest of data? Please, consider revision.
Response: The unit of all data in Table 2 is “μmol Trolox equivalent per gram of sample”. We have revised Table 2.
- Finally, I encourage authors to perform a revision of the whole manuscript by a native English speaker. Several parts are not well understood and it could be due to the language.
Response: Thank you for your suggestion. We have revised other parts of the manuscript.
Reviewer 3 Report
The present study highlights the influence of L-Cysteine on Maillard reaction by limit/ reduce the content of 5-hydroxymethylfurfural. The proposed mechanism of reaction and formation of DTH compound can potentially contribute to the elimination of harmful and toxic Maillard-derived compounds from temperature-treated food samples. Nevertheless, a few manuscript’s aspects should be taken into account to fulfil the Journal’s requirements, as described below.
- Abstract. "Maillard reaction (MR) lowers the nutrient utilization and produces harmful pollutants." I disagree with this statement because during MR antioxidant compounds and compounds responsible for the positive aroma and colour features could be also formed, therefore I propose to change it to "Maillard reaction (MR) could lower the nutrient utilization and produces harmful pollutants."
- DTH abbreviation in the abstract should be written in full form.
- Line 33-35: Please also consider here that brown pigments are also melanoidins, which has positive antioxidant potential and influence on creating for example the brown colour of bread crust.
- Line 69: all the sulfhydryl compounds used in the study should be mentioned here.
- There are 9 references which were published before year 2000, please try to change them to more up-to-date literature.
Author Response
Itemized list of responses to reviewers’ comments
Comments of Reviewer 3:
- Abstract. "Maillard reaction (MR) lowers the nutrient utilization and produces harmful pollutants." I disagree with this statement because during MR antioxidant compounds and compounds responsible for the positive aroma and colour features could be also formed, therefore I propose to change it to "Maillard reaction (MR) could lower the nutrient utilization and produces harmful pollutants."
Response: Thank you for your suggestion. We have revised it in Line 14-15.
- DTH abbreviation in the abstract should be written in full form.
Response: Thank you for the suggestion. We have revised it in Line 20-21.
- Line 33-35: Please also consider here that brown pigments are also melanoidins, which has positive antioxidant potential and influence on creating for example the brown colour of bread crust.
Response: Thank you for your suggestion. We have revised it in Line 37-39.
- Line 69: all the sulfhydryl compounds used in the study should be mentioned here.
Response: Thank you for your suggestion. We have revised it in Line70-73.
- There are 9 references which were published before year 2000, please try to change them to more up-to-date literature.
Response: Thank you for your suggestion. We changed part of the 9 references. But reference [13], [31] and [32] were not changed because they are important and represented the early research in corresponding field.
Round 2
Reviewer 1 Report
Line 36-38: in the revised version of the paper, authors still describe Maillard Reaction Products as noxious substances for health. It was asked to introduce both the positive and negative aspects of the reaction before focusing onto food contaminants. Please, add at least a sentence to describe that.
Line 174: Authors have added in their methods part information about the preparation of a standard curve for DCH. They, however, do not mention how they got the standard and how they assessed its purity.
Line 193 (Cytotoxicity assay): Please mention in the text the various concentrations of DCH and HMF used for the cytotoxicity test.
Author Response
Comments of Reviewer 1:
- Line 36-38: in the revised version of the paper, authors still describe Maillard Reaction Products as noxious substances for health. It was asked to introduce both the positive and negative aspects of the reaction before focusing onto food contaminants. Please, add at least a sentence to describe that.
Response: Thank you for your suggestion. We added some sentences “MR produces many flavor compounds such as aldehydes, ketones, pyridines, and pyra-zine, which form the special flavor of some foods [1]. Especially for coffee and bread, MR provides tantalizing aroma and color to increase customers' appetite.” in Line 34 -37.
- Line 174: Authors have added in their methods part information about the preparation of a standard curve for DCH. They, however, do not mention how they got the standard and how they assessed its purity.
Response: Thank you for your comment. Due to the lack of commercial standard DCH, we decided to prepare the standard by ourselves. The standard DCH was isolated and purified using macroporous resin HP-20 and sephadex LH-20 in Method 2.4. No impurity was found in the product by 1H NMR spectrum (Figure S4). We revised it in Line 179-180.
- Line 193 (Cytotoxicity assay): Please mention in the text the various concentrations of DCH and HMF used for the cytotoxicity test.
Response: Thank you for your suggestion. We added” The concentrations of HMF or DCH were prepared as 80, 160 and 320 μM” in Line 195-196.
Reviewer 2 Report
All suggestions and scientific questions have been properly addressed by authors.
Author Response
Thank you for your great help